# Data Evidence-Based Transformative Actions in Historic Urban Context—The Bologna University Area Case Study

**Saveria Olga Murielle Boulanger, Danila Longo**  **and Rossella Roversi ***

Department of Architecture, Bologna University, Viale del Risorgimento 2, 40136 Bologna, Italy;
saveria.boulanger@unibo.it (S.O.M.B.); danila.longo@unibo.it (D.L.)

* Correspondence: rossella.roversi@unibo.it

**Abstract:** The rapidly growing use of digital technologies in urban contexts is generating a huge and increasing amount of data, providing real-time information about the urban environment and its inhabitants. The unprecedented availability of data allows us to not only improve advanced knowledge and gain a deeper understanding of urban dynamics, but also enact data evidence-based transformative processes and actions in the direction of smarter, more sustainable, resilient, and socially equitable cities. In this context, the literature on smart cities has recently expressed the need to more deeply involve urban visions and communities in the process of regeneration. This paper aims to analyze how big data can be useful in understanding the effectiveness of small pilot actions of regeneration and reactivation in valuable cultural heritage (CH) urban environments. Pilot actions were developed in the context of the European Union funded project "ROCK—Regeneration and Optimization of cultural heritage in Creative and Knowledge cities" (GA730280). The paper analyses data collected by the ROCK City People Flow tool, in different use and time conditions, in two central squares of Bologna (Italy), in order to rate event successes, spatial transformation effects, and regeneration tactics responses. Data confirm the complexity of interpreting phenomena in such contexts but also provide useful indications for future planning.

**Keywords:** cultural heritage; sensors; crowd monitoring; urban regeneration; big data; transformative urban action; People Flow tool; smart city; historic urban context

## 1. Introduction

The rapidly growing use of digital technologies in urban contexts is generating a huge and increasing amount of data, providing real-time information about the urban environment and its inhabitants. To deploy the potential of this data deluge, the data format must be organized, elaborated, made homogeneous and comparable, interpreted, and analyzed with the aim of enhancing the knowledge of urban phenomena. Decision makers, policymakers, service providers, urban planners, and designers have at their disposal instant views of various aspects that compose urban complexity and have the opportunity to utilize them to develop new forms of governance, functioning, and planning.

The unprecedented availability of data allows us to not only improve advanced knowledge and gain a deeper understanding of urban dynamics, but also to develop data evidence-based transformative processes and actions in the direction of smarter, more sustainable, resilient, and socially equitable cities: "overall, the new era of science and technology embodies an unprecedentedly transformative and constitutive power—manifested not only in the form of revolutionizing science and transforming knowledge, but also in advancing social practices, producing new discourses, catalyzing major shifts, and fostering societal transitions" [1] (p. 3).

Traced by a variety of tools and techniques, citizens may generate information about their movements in their spatial, geographical, and temporal contexts, manifesting needs, expressing preferences, and giving feedback and responses to public and private initiatives [2,3]. The collected datasets about urban systems and their inhabitants, if adequately harnessed and processed, can reflect the way citizens use and experience urban spaces, services, and infrastructures, and the way cities react to climate change. This kind of data application is important not only to comprehend the real-time situation under investigation, but also to generate useful knowledge to predict future dynamics and to enhance decision making. It can be transformed in a tool able to enhance the comprehension of the impacts of implemented transformative actions and strategies in order to verify their effectiveness and support their design. Such feedback is valuable for all levels of city planning and governance. Additionally, when this approach is applied to pre-existing built environments characterized by the presence of cultural heritage (CH), the availability of data on citizen behaviors related with urban space can provide researchers new ways of designing and understanding the most efficient transformations [4]. We are in fact living an era when urban contexts often need to understand ways to improve their resilience, mitigation, and adaptation to climate change, in a way that is also in line with citizens' needs [5] (p. 16).

This study deals with the research experience of a design data-guided approach nurtured during the "ROCK—Regeneration and Optimization of Cultural heritage in Creative and Knowledge cities" project [6]. ROCK is an EU-funded project, under the Horizon 2020 program and under the call SC5-21-2016-2017 "cultural heritage as a driver for sustainable growth" (GA 730280). The three-year project, which is now nearing its end, is coordinated by Bologna Municipality with the technical and scientific support of the University of Bologna. The context of the research is historic city centers and their tangible (buildings, monuments, squares, and streets) and intangible (creativity, art, productions and services, and people skills and abilities) cultural heritage [7,8]. CH is not simply considered a valuable asset to be preserved, but a privileged field for inquiries and actions that may transform city centers into laboratories in order to test new models of urban regeneration, sustainable development, and economic growth [9]. ROCK aims to test actions, strategies, and tools to foster the usability of public spaces to all, improving CH functions from a user perspective, defining key policy issues, integrating emerging spatial, temporal, and virtual structures of the knowledge-based society to support social cohesion and foster the involvement of communities in the process of re-appropriation of neglected or misused public space. ROCK supported the transformation of significant urban areas in three historic city centers, situated in Bologna, Lisbon, and Skopje (replicator cities) into creative and sustainable districts, by implementing a repertoire of successful heritage-led regeneration initiatives already experimented in seven selected cities (Athens, Cluj-Napoca, Eindhoven, Liverpool, Lyon, Turin, and Vilnius, called role-model cities) through a continuous knowledge–experience exchange process. The replication potential and effectiveness of the tools and policies developed in the role model cities are tested in the three replicator cities, taking their specific historic context and local needs into account.

To answer the ROCK question on how it is possible to convert historical cities into smart, resilient, sustainable, creative, and knowledge cities, the development and adoption of tools and technologies is a key aspect [10]. The experience of the ROCK project shows how orienting and equipping urban planners with specific tools for measuring the impact of actions and policies related to the new role of the CH is a particularly relevant issue [11].

For this reason, the paper focuses in particular on the results of a large-crowd monitoring tool in relation to temporary pilot urban transformations that have been implemented in the historic city center of Bologna, Italy, during the ROCK project. The objective of the paper is to analyze the presence of people in public spaces and their variation according with different situations, in order to rate spatial transformations effects and regeneration tactics responses. The availability of a big dataset allows us to verify the impacts of the adopted measures and provides useful elements to facilitate the planning of new ones.

The research question underlying the paper is the following: how useful can data about people presence in existing urban areas be for understanding how to address urban modifications and transformations, especially when it comes to cultural heritage?

The paper is organized as follows: the first section intends to briefly relate this paper with the major state of the art work on the smart city topic; then the second part introduces and describes the selected case study in the framework of ROCK project; the third part identifies the methods which were used for the analysis, and a brief description of the technologies beneath the used sensors, as well as some limitations and the main method we used for analyzing data; the fourth part shows the results of the analysis and their discussion. In conclusion, we address the core findings of this study, linking them with the current situation of the COVID-19 emergency as well, and how both data and temporary transformation can support a deeper use of open spaces inside existing urban areas.

*State of the Research: Among Smart Cities, Cultural Heritage-Led Regeneration and Big-Data*

According to Vallianatos (2015) [12] and Caragliu and Del Bo (2018) [13], the concept of smart cities emerged in the second part of the last century, probably around the 1960s, when first attempts on computer databases, cluster analysis, and aerial photography were done in the Community Analysis Bureau. Since then, the concept has seen a growing interest, appearing again in the 1980s and then more frequently by the 2000s, when it was discussed in an international debate (Komninos, Mora, 2018).

Since the beginning, some elements appeared to be interlinked: the presence of innovative technologies, which provided innovative ways for understanding the world, the presence of a growing digital scenario where ubiquitous technologies allowed several services to be detached by physical spaces and, finally, the city as the main potentially evolving environment, where those technologies should take place. According to Komninos and Mora 2018 [14], in fact, in these years several definitions emerged, such as "digital city", "ubiquitous city", "cyber city", and similar: "in the early writings, the demarcation lines between these concepts were fuzzy and all these terms attempted to capture the same information-based and knowledge-driven development process of cities" (p.18). Hence, since the beginning of smart cities concept, technologies and innovations were already linked with the city as the main object and the localization of those innovations.

The concept of smart cities in the following years, until today, was slightly modified and specified in relation to several factors, but mainly according to two core tendencies: on one hand, several researchers tend to focus on the "hard" part of smart cities technologies; on the other hand a growing number of researchers invite reflection on the "soft" parts, mainly identified in the role of citizens in the process, but also in the difference between top-down and bottom-up approaches, in the presence of several ways of financing and, in conclusion, in the need to identify impacts and reliability of methods to measure the results as well as the effectiveness of the approach [14] (p. 29), [15] (pp. 28–30).

Parallelly, several researchers started to work in the direction of understanding how smart cities should be aligned with the current growing urban challenges. According to them, in fact, smart cities should answer the most pressing urban issues with the use of technologies [16,17]. Among these challenges, adaptation, mitigation to climate change, and resilience are included along with inclusion, equity, and poverty, as stated in the Sustainable Development Goals (SDGs), especially in numbers 11, 3, 4, 5, and 10. It is recognized by the worldwide debate that technologies can play an important role in shaping the cities of the future and, especially, in supporting addressing several existing issues of mitigation and adaptation to climate change, poverty, equity, security, etc. [18–25]. The same definition of smart cities, given in 2014 by the Directorate General for internal policies, claims the following aspects: "a Smart City is a city seeking to address public issues via ICT-based solutions on the basis of a multi-stakeholders, municipally based partnership" [26]. However, it is crucial to remember the role of technologies in the definition of urban futures. As recently expressed by Ben Green [27], technology must not be the aim of the smart city, but the means to achieve the vision that each city has for itself. This concept is not new. Boulanger (2020) [15] stated as well that the smart city needs to apply technologies in order to reach higher goals, especially the ones that try to solve the big problems which

society is currently encountering, such as climate change, poverty, climate migration, social equity, access to basic services, etc. In fact, in both Green and Boulanger, the proposition is to push cities to frame their vision before choosing which technology should be implemented. Indeed, there are several studies focusing on the role of citizens and communities in this context. In fact, citizens and their lives need to be the primary objectives for cities. As a matter of fact, one of the earliest definitions of smart city, created by some of the first theorists of this topic, Giffinger and Fertner, says: "a smart city is a city well performing in a forward-looking way in these six characteristics, built on the smart combination of endowments and activities of self-decisive, independent and aware citizens" [28,29].

As it is possible to read in the Sustainable Development Goal number 11: "cities are hubs for ideas, commerce, culture, science, productivity, social, human and economic development. Urban planning, transport systems, water, sanitation, waste management, disaster risk reduction, access to information, education and capacity-building are all relevant issues to sustainable urban development". Cities are, in fact, seen as complex environment, composed by several layers and framed by a multiplicity of challenges that need to be addressed by services, projects, and solutions [30]. The use of technology is one of them, together with nature-based solutions, temporary transformations, and others. Their application in the management of the urban project is less investigated at district scale than at the scale of the individual building [31].

When it comes to understanding how to solve current challenges in existing urban areas, especially if framed by CH, it is interesting to understand how technologies can be mixed with other solutions. As stated by the European Commission in a recent publication written by the high-level expert group on innovating cities, "digitisation, designed and used well, can help create a fourth industrial revolution that is lean, clean and green" [32] (p. 14). In particular, when thinking about CH in existing urban areas, some of the most interesting approaches deal with the idea of adaptive reuse. As stated in the Lewarden Declaration [33] (2018), "adaptive re-use requires the adoption of a 'living' attitude vis-à-vis our built environment; an attitude that considers our built heritage as a man-made landscape that can be re- worked and re-modelled when necessary, starting out from the social, cultural, environmental and economic needs of our time. In so doing, our built heritage can be integrated in a meaningful and creative way into contemporary society and thereby be conserved in a sustainable way for future generations". In the idea of Barbara Camocini, the first architect to define the adaptive reuse locution, cultural heritage needs to be considered not only as a good to be preserved, but also mainly as a resource to be included in the strategies for solving the most pressing challenges, for example, degradation, lack of sense of belonging to spaces, etc. [34]. CH has always been adapted and transformed across ages in relation with the evolving urban necessities [35], and, following this line, the European Commission also identifies CH as a means for achieving a high level of urban regeneration.

Cultural heritage has been actively supported by the European Commission for years. In fact, several work programs have focused on that field of research as a core element able to create a sense of citizenship and belonging to space and cultures, thus making citizens and communities more aware of current problems, such as social inclusion, climate change, ethics, racism, and acceptance.

Indeed, research on CH has been consistently supported within EU framework programs: projects related to the main aspects of CH, including digital related issues, under the Seventh Framework Program for research and technological development have been funded with around €200 million and under Horizon 2020, the almost-concluded EU framework program for research and innovation for 2014–2020, with estimated €500 million [36]. During the presentations of the new Green Deal calls held last September and October, the EU anticipated that CH would still be present, and it would represent a key aspect in the next Horizon Europe program as well.

Both in smart cities and CH studies, citizens and communities are perceived as core parts of the discussion for addressing changes in the already-existing urban contest. As stated in the last European Commission presentation of the new Green Deal calls by John Bell (Director of the Healthy Planet Directorate—DG Research and Innovation), it is not possible to achieve the objectives, which was set up by the Paris Agreement and included into the Green Deal, without people and without deeply

involving them in each action made in cities. Furthermore, as also stated in the human-centered city, 2019 report from the European Commission, "cities should be more human centered. This is not just a right, but also involves responsibilities, obligations and duties. To be a city for citizens where citizens become city-makers and shapers, makers and co-creators of their evolving urban development is not an entitlement. It means being an active citizen concerned with the local and European context and with the urgency of the global context" [5] (p. 16).

According to these concepts, the ROCK project defined its activities and actions. In fact, even before thinking about which, where, and when applying technology, the project supported the cities involved (Bologna, Lisbon, and Skopje) in thinking about their future and their visions (called "scenarios" in ROCK) for creating a regeneration led by tangible and intangible CH (for further information, check the ROCK website where several deliverables are available on that topic).

Even with the limitations expressed before, technologies can effectively play an important role in the understanding of people behavior, especially when it comes to understanding the way in which they use the space, and if temporary transformations are successful or they need to be adapted to different uses. The role of temporary transformations and small interventions is perceived by many authors as an interesting alternative to larger scale actions [37]. This is because doing actions in the form of acupuncture [38,39] seems to be more efficient, less time consuming, less expensive, and more flexible to future modifications. Going further and as stated by Iaconesi and Persico: "Digital Urban Acupuncture (DUA) is a participatory urban intervention methodology and practice in which knowledge of the Relational Ecosystem of cities—including the flows of communication, information, knowledge, emotion, expression, the communities in which people gather, and the roles which people play in them—is diffused and made available to all stakeholders using artistic, creative ways—such as exhibits, museums, visualizations, performances, workshops—so that they can learn how to understand it and its implications, and how to use it to achieve their goals by using the relational networks to create operative, performative connections and, thus, (re-)establish flows to "make things happen" in the city" [40].

Indeed, all these forms of smaller actions can be easily supported by digital technologies, not only in their creation but mainly in the monitoring of the effects that they produce. The ROCK project, in this regard, decided to implement people flow analytics (which is the core of this contribution) but also other types of sensors, such as climatic together with a sentiment analysis based on social media feedback and press articles [41]. However, this paper will mainly focus on people flow as a first way to understand the effectiveness of these typologies of actions in the specific case of the city of Bologna.

The analysis of CH-related research works, limited to the covering of those involving the applications to the Internet of Things (IoT), linked with the core enabling technologies of big data analytics or applied technologies, highlights a concentration of studies about methods and solutions to face and protect CH sites from threats using non-invasive technologies; refers to digital opportunities for reconstruction, re-creation, and co-creation; and is about immersive experiencing of cultural heritage and about methodologies, devices, strategies and systems for monitoring, and assessing cultural heritage under different conditions, in particular against climate change and disasters [42,43]. An example is the EU "STORM—Safeguarding cultural heritage through technical and organizational resources management" [44]; it is an H2020 funded project that aims to develop a toolkit of technologies and low-cost methods to protect CH from threats and to predict potentially dangerous environmental changes. STORM uses eco-innovative, non-invasive, non-destructive sensors and solutions to perform surveys or real-time monitoring on heritage assets, including "human sensors" through crowdsensing and crowdsourcing techniques [45].

The state of the art about real-time on-site CH monitoring and about the collection and processing of data coming from different sources, including sensors [46], shows that the studies are mainly devoted to supporting decision-making processes related to safety and security, considering all phases in the disaster lifecycle (prevention, preparedness, response, and recovery activities) [47–49] or supporting the assessment of the structural health of buildings and monuments [50]. The application of sensors

is considered a crucial aspect for the study and diagnosis of artworks and artefact, especially for diagnostic purposes [51–53].

There are only a few studies that have recently focused on the use of big data applications in support of decision-making processes about management, governance, and data-driven urban design in CH context; of course, most of the research concerning urban issues is related to smart city domains [54]. Nevertheless, European historic city centers, representing more rigid and delicate contexts and presenting low propensity to change, need carefully balanced actions to maintain an equilibrium between valorization and conservation. Data help to monitor and investigate city life and environment changes over time, to build predictive models concerning city development and planning, and to check the outcomes and results of initiatives and actions, all based on rational evidence.

For this reason, the ROCK project used video neuro-analytics (VN) and city people flow applications to determine public response to heritage-led regeneration actions. Combining anonymized data analytics on people's emotional states and environmental data—such as air temperature, humidity, wind, air pollutants—it was possible to infer how public spaces are used and perceived. In Elisei, P. et al. [11], the results of the large-crowd monitoring application in two ROCK role model cities, Cluji-Napoca (Romania) and Turin (Italy), are illustrated. The data collection and interpretation aimed to analyze and correlate people flows and behaviors to specific periods in order to identify impacts of initiatives carried on in selected areas.

The present paper deals with another ROCK application of the City People Flow tool [55] to detect the presence of people in public spaces in the Bologna historic university area (so to rate event successes, spatial transformations effects, and regeneration tactics responses). The objective is to demonstrate that data-driven initial evaluations (before implementing the action), monitoring (during the action), and feedback (after the action, if temporary, as those explored by the present contribution) are crucial to steer urban strategies and support decision-making addressing regeneration and re-activation in CH contexts, where the involved connatural limitations and barriers might lead to inaction or failed experiments.

## 2. Case Study Selection in the ROCK Framework

Before going specifically into the methodology used for the analysis, it is important to present the case study we selected because of its specific relations with the obtained results, and it is particularly interesting in relation with the research question. As the research question deals with the comprehension of technologies and big datasets related to people which can be useful for understanding how to address urban transformations in existing urban areas, the case study of Via Zamboni seemed very interesting for several reasons: at first, within the ROCK project, we developed a methodology of research-action-research (see Section 2.1) for the area that led to the creation of pilot interventions in the form of physical transformations (see Section 2.2) in combination with events and other initiatives; then, those physical transformations were co-designed with the community; finally, they were an example of adaptive reuse of space framed by the presence of CH. Thus, the case study of Via Zamboni seemed to be in line with the research we wanted to perform.

However, as detailed in Sections 5 and 6, according to this paper, CH is not a variable of the methodology, but it is a condition of the case study that needs to be taken into account for the discussion. In other words, the present paper does not mean to investigate whether the temporary transformations were effective in relation with conservation or enhancement of cultural heritage, but it takes this point as a feature of the area considered.

The following section describes the case study and the physical transformations in the Via Zamboni area as a result of ROCK project and its methodologies. This description is necessary to highlight the complexity of the area and the interest in using sensors and a big dataset for understanding the success or failure of those typologies of actions.

*2.1. Research-Action-Research Approach: The ROCK Method for Implementing Actions*

ROCK methodology is based on an experimental pilot process, driven by a research-action-research sequence, intended as an iterative process working according to the following scheme [56]:

1.  Research. The first research phase aims to create the knowledge base and outline the initial status by collecting data and needs, and identifying key stakeholders, key areas, and key actions and enablers, in order to effectively set up the field for the concrete planning and implementation phase.
2.  Action. The action phase concerns the implementation of pilot actions according to the first drafting of objectives, topics, and proposals emerged from participatory inquiries. This first action can highlight unforeseen barriers, arise more needs, or suggest additional adjusting actions. Therefore, the process foresees a second research activity.
3.  Research. The second research phase aims to define more precise and detailed scenarios, which includes not only foreseen actions and tools application, recalibrating the future actions to improve their effectiveness, strengthening week aspects and correcting mistakes, but also considerations about new assumptions to be taken into account, new barriers and risks, new stakeholders, and the connection and clustering of specific actions with new or more precisely identified target groups.

The methodology is intended at the urban scale but, given the limited duration of the project, ROCK implemented several pilot actions at the district/neighborhood scale in all the involved cities. This smaller scale experimentation provided an intermediate dimension useful for effectively carrying out projects and monitoring results.

According to the ROCK research-action-research iterative approach, monitoring and dataset availability are strategical because part of each phase (a reliable initial state, the prediction of the desired future scenario, the monitoring of the ongoing phases of the implementation, the collection of feedback from people and environment, the reading and elaboration of the results) may take great advantage from the deployment and implementation of sensor technologies and networks. More insights into the project methodology are available on the project website [6].

*2.2. Case Study Selection*

Bologna is one of the three replicator cities of the ROCK project, selected for its role as a knowledge-based city thanks to the presence of the oldest European University, the inclusion in the UNESCO Creative Cities' List, and the efforts of the municipality and the city community in regenerative practices and a collaborative approach. In the city of Bologna, the ROCK project has been able to install a high number of sensors in the area of Via Zamboni. Bologna decided to highly invest in using data from sensors in order to understand the effectiveness of actions in the area, while the other two replicator cities decided to focus more on other aspects or on different tools.

The area of Via Zamboni, which is mainly a pedestrian area (only a few authorized people can access by car and only for very specific reasons), is of particular interest, in order to understand how technology can contribute to defining ways to regenerate existing urban areas characterized by the presence of tangible and intangible cultural heritage. This is because the area contains both elements, being the (traditional) core of the university and being located inside the most historical and more ancient area of the city. There, there is a variety of buildings and squares protected for their artistical and testimonial role in the development of the city. Additionally, the presence of the university makes the area crowded by students, professors, and university staff for most of the year. Moreover, the street is also frequented by young people, in general going there for the presence of pubs and restaurants; by artists and citizens going to the main city theatre; by residents; by people that simply pass there; and finally, by tourists. The street is also, in fact, one of the three main streets connecting the Two Towers square with the rest of the city and it hosts some interesting museums.

However, the area has seen in history several episodes and moments of decay and carelessness, despite the efforts made by different administrations. It has been the place for protests and, in general,

it has been perceived as a degraded area of the city center. The presence of micro-criminality and trafficking together with the noise linked with the nightlife that usually comes from that area has increased this perception.

Piazza Verdi is the central, biggest, and most important square facing Via Zamboni. There is located one of the most important cultural institutions of the city, the Bologna Municipal Theatre, but it represents also the day- and night-time gathering place for students and marginalized sectors of the population, so causing conflicts with the residents, vandalism, scarce safety perception, and under-estimation of CH.

It is also for those reasons that the ROCK project decided to implement actions there, in order to enhance both the official and non-official culture present in the area through a variety of temporary and permanent actions. In particular, the project decided not to implement a unique big action, but a set of multiple and linked smaller actions, to make the multiple under-used squares along Via Zamboni new-lived urban spaces. The attention has been, for that reason, concentrated in testing different pilot actions in the different squares, also according to the vocation and the characteristics of each one.

This paper intends to address two correlated types of these actions: the implementation of temporary physical transformation of outdoor public spaces together with the monitoring through big data of those actions.

Temporary Spatial Transformations

Among the pilot temporary transformations implemented in Bologna city university area, the present contribution describes the impacts of actions that took place in two important squares located along Via Zamboni, Piazza Scaravilli and Piazza Rossini.

Piazza Scaravilli, characterized by the presence of porticoes on its four sides, despite the location in the core of the university area, has always been underused by students and citizens because of the lack of equipment and "urban atmosphere", and, on the other hand, the presence of a car park. Thanks to Malerbe temporary transformative action, Piazza Scaravilli has been changed into a dynamic garden, where cultural events, aggregation, and entertainment can be carried out, for the re-appropriation and the continuative use of space. The objective of giving life to this neglected area is matched with the objectives of mitigation of climatic effects. A participatory design process led to the definition of the temporary architectures, composed by light, modular, and reversible elements, that was co-constructed by students and citizens (Figure 1).

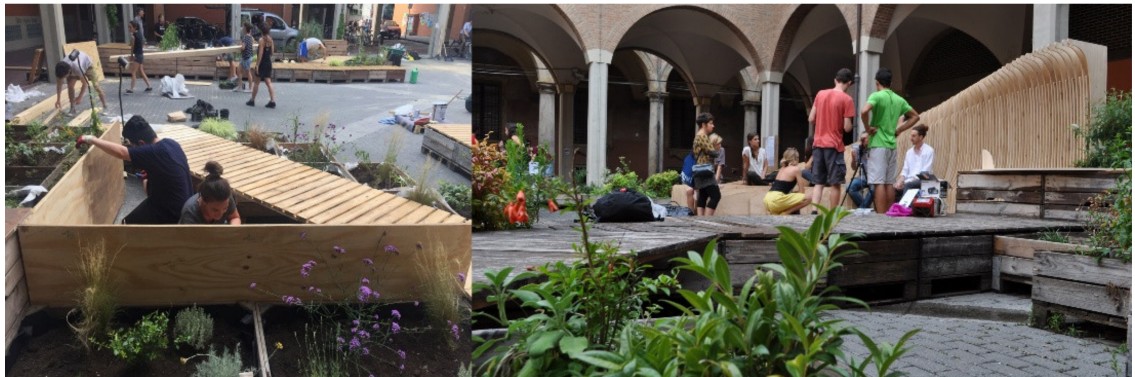

**Figure 1.** Wooden elements: seats and planters in Piazza Scaravilli during a Malerbe co-construction session (credit: Regeneration and Optimization of cultural heritage in Creative and Knowledge cities (ROCK) project).

Another temporary action implemented in Piazza Scaravilli is "Screensaver". The experimentation was part of a broader strategy of actions that involved all the main squares along Via Zamboni—as recalled by the name of the initiative, "Le Cinque Piazze—The five squares"—developed in synergy with the Bologna Design Week and Researchers' Night in September 2019. The objectives of the design

proposals and the related events were the enhancement of CH and its accessibility through imagining and testing different and unconventional uses, according to an environmental sustainability perspective.

The temporary design actions provided in Piazza Rossini, implemented through two main and successive phases of transition, led to the permanent transformation path of the square into a new pedestrian and regenerated space for both citizens and city users. Piazza Giacomo Rossini is a public space of relevant historical and architectural value, overlooked by important private and public institutions for which it has always served as a parking lot. "Green Please: the meadow you don't expect" is the first temporary project for the spatial transformation and the use redefinition of Piazza Rossini square, and the result of a shared and participatory process, part of "Le Cinque Piazze" initiative. The temporary project transformed Piazza Rossini into a green space, proposing an unexpected perception of the area and introducing the possibility to stay, appreciate the view of the surrounding CH, and enjoy the events that the new configuration of the square allows (Figure 2).

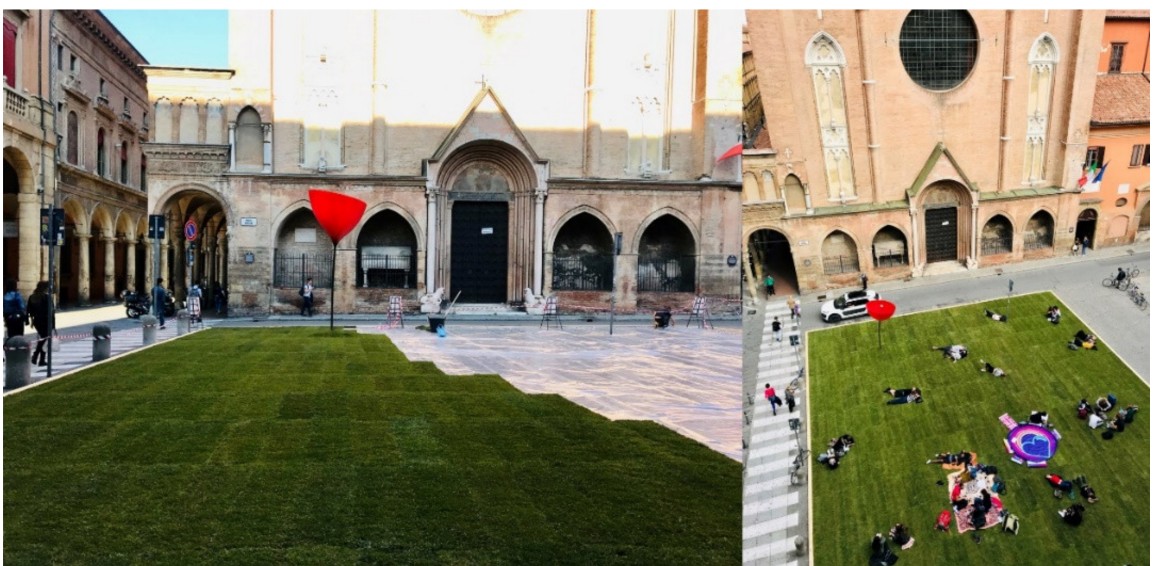

**Figure 2.** On the left, a phase of the making of the "Green Please: the meadow you don't expect" project; on the right, one of the possible ways citizens and visitors use the square (credits: ROCK project).

The success of the initiative, confirmed by a citizen's signature collection asking for the definitive pedestrianization, enthusiastically endorsed it as the first step towards a different future vision of public spaces, potentially extensible to all the city. To cover the period needed for the definition of the permanent layout of the square, a new temporary transition project was designed: "Green Please 2.0! The green you don't expect. A project for piazza Rossini in transition", implemented and realized during the 2020 summer. The new square configuration, currently ongoing, is conceived as a development of the previous one since its longer stay implies design and technological solutions of a higher level of complexity. In addition, the project has educational, social, and awareness purposes on ecological issues, environmental, and common CH care (Figure 3).

It must be underlined that, as the further sections will highlight, the above-described spatial transformations open the areas to different uses, including the hosting of events, so it is important to have feedback on the success of the initiatives and check the effectiveness of the designed solutions, in order to confirm the expected performance or to set up optimization and corrections.

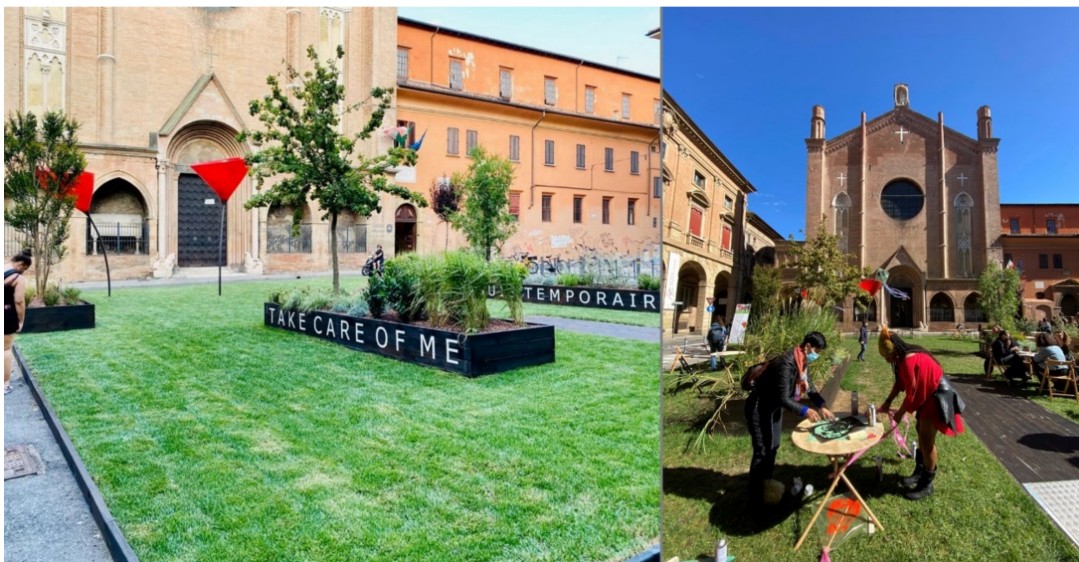

**Figure 3.** On the left, "Green Please 2.0! The green you don't expect. A project for piazza Rossini in transition"; on the right, activities during an event in October 2020 (credits: ROCK project).

## 3. Methodologies

This section deals with the methods used for the analysis presented in this paper and addressed to answer the research question dealing with understanding how big datasets from sensors can be useful for assessing the success of physical transformations in already-existing urban areas. The paragraph starts with the description of the used sensors and their position in the area, it shows some limitations about the data coming from those sensors, it describes the different variables we considered for the analysis, and, finally, it specifies the different steps that the study has undertaken.

### 3.1. The Use of Sensors in the ROCK Project in Bologna

As highlighted inside the introduction, it is interesting to understand how data obtained by large crowd monitoring sensors can help in determining if the temporary transformations were successful in making neglected/misused spaces more usable and friendly for people's lives and activities, and if they were able to attract people in the area, making its CH more "known", accessible, and explored.

We used data coming from ROCK tools, in particular, the data coming from a set of sensors deployed in the city of Bologna by the project technological partner Data Fusion Research Centre (DFRC) [55]. The partner installed 10 sensors along Via Zamboni (Figure 4) in collaboration with the Municipality of Bologna and the University of Bologna (both project partners). All installations received the authorization from the Soprintendenza di Bologna, the local agency Italian public body that has the role of CH protection.

The sensors collect data in real-time through the technology WiFi/GPS e 4 kit. Data are then collected and shared, through personal and secret credentials, with selected project partners only in an aggregated form (so that it is not possible to identify single persons). The sensors detect the passage of people by intercepting, in an anonymous and aggregated way, the WiFi or GPS signal of smartphones. The data collected concern the number of people (devices) present in an area within a radius of about 50 m from the sensor, their movements within the covered area, and the nationalities of those present.

The data in an aggregated form are then uploaded in open access into the ROCK Interoperable Platform, developed and managed by Corvallis [57]. The platform allows downloading data in an Excel format, or it is possible to query them through an open data dashboard [58]. The dashboard allows querying data in relation with time, type of sensor, location of the sensor, and city.

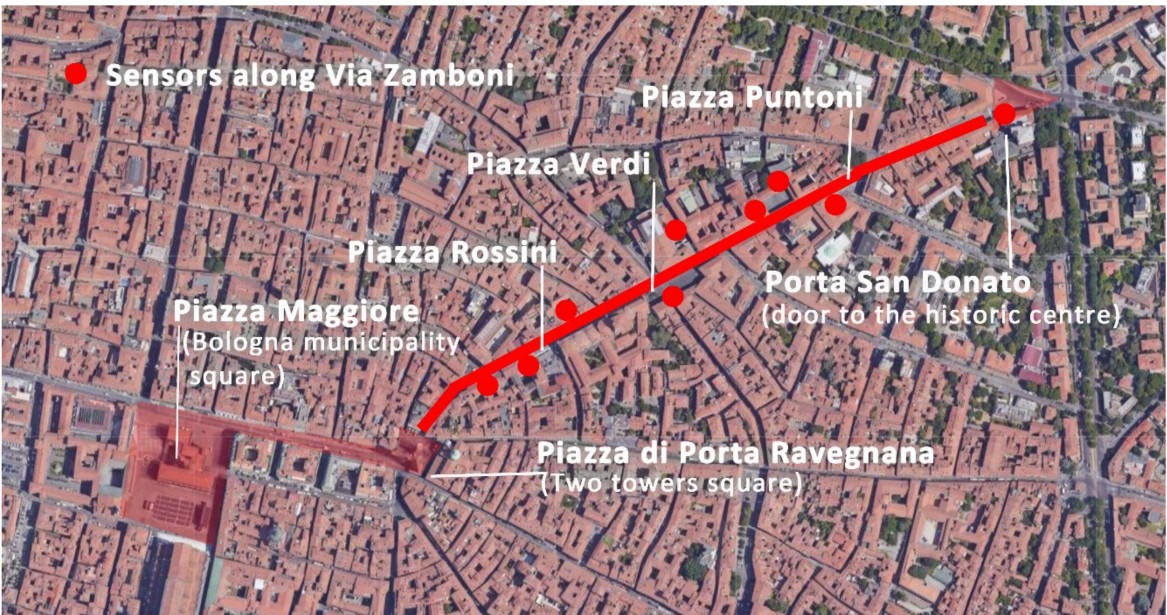

**Figure 4.** The sensors for large crowd monitoring positioned along Via Zamboni in Bologna.

3.1.1. Limitations on the Sensors in the Area

Before investigating the methods we used for analyzing data from those sensors, we need to acknowledge some limitations in the installation of sensors in the area, as detailed following:

- Geographical limits. The sensors were installed along the street named Via Zamboni according to several boundary conditions, such as the availability of physical space on building façades, the proximity with electrical connection points, the different ownership of buildings, and the needs of the Soprintenza ai Beni Culturali. All those boundary conditions made the selection of the best position for the sensors limited to certain specific spots. For that reason, the municipality and ROCK partners were not able to always place sensors in the best potential spot. In the case of Piazza Rossini, for example, the sensor can detect not only people entering the square but also the ones passing in front of it, causing some interpretation limits. In this study, we acknowledge those limits, and we consider them in the interpretation, as explained in the results and discussion section.
- Time limits and unpredictable events. For the reasons explained in the first point, the time needed for completing the installation was higher than expected. In fact, the municipality and ROCK partners were able to make sensors in function starting from June 2019. The process of acquiring authorization, doing in site inspections, solving technical issues and, finally, the need to address ethical issues, as requested by the European Commission, led to a delay in the installation. Thus, we are not able to compare 2019 results with 2018. Additionally, the emergence of COVID-19 caused a sensor switch-off in the entire area, due to the temporary closing of the partner in charge of them. Finally, the sensor in Piazza Scaravilli encountered some technical issues, thus we have limited availability of data from this site (from November 2019 to July 2020 no data are available). We included the specifications of that in the results and discussion section. However, the project planned to maintain the monitoring active for two years after the project end (December 2020), thus we aim to continue the analysis with these big data. However, even with these limitations, data show interesting features that need to be addressed and taken into consideration.

*3.2. Data Analysis Methodology*

The method used for analyzing data is directly linked with the two platforms where data are made available: the CKAN platform, an open source data management system, with business analytics section available at the opendata.rockproject.eu, and the DFRC business analytics platform which

is inside the analytics.lbasense.com platform called LBAsense. Data from June 2019 to July 2020 were available in the ROCK open data platform, while August and September 2020 were available in the LBAsense platform. Each of these platforms allows data to be queried in different ways: time range, time grain, geographical area. From the query, several graphs are automatically made available: average daily visitors per region, sum daily visitors per region, sum daily visitor's timeline, daily people average per duration, and temperature linked with footfall. Data were available from June 2019 and September 2020. Thus, we decided to consider this entire timeframe.

For the purpose of this paper, we decided to perform the following analysis:

- The first one dealt with analyzing the total and the average number of people detected by sensors on a monthly basis for the entire available timeframe in the entire area of Via Zamboni and in the two squares of Piazza Scaravillli and Piazza Rossini. This analysis intended to understand the general frequency and amount of people in the areas and to find peaks.
- The second one dealt with analyzing the total and the average number of people detected by sensors in very specific timeframes of eight days each (see Section 3.2.1 for further information on how we chose the timeframes) in the same three areas. This analysis aimed to understand the potential correlations among peaks and the presence of events or physical transformations in the area.
- The third one dealt with understanding the distribution of people in the area. We performed this analysis only for Via Zamboni, in order to understand if there was some variability in relation to the different timeframes.
- Then, we addressed the duration of people in the two squares. This analysis was particularly useful to understand how people tend to use the space and to identify how many people just pass the area and are detected by the sensors, and the ones that have an interest in the specific square and thus they stay longer than 5 min. For this purpose, we found different durations as explained in Table 1.
- Finally, we studied the concentration of people in the two squares along the day, to understand the most frequented hours. For this analysis we considered the entire 24 h of the day, to analyze not only the daytime but also the nighttime.

**Table 1.** Interpretation of the different durations of people stay in the area of Via Zamboni.

| | |
|---|---|
| <5 min | People passing in the area without stopping. These are people that use the street as a means for transit from one side to another |
| 5 to 20 min | People stopping for short time in the area, for short purposes. These can be variable, e.g., from short activities (like taking a coffee), to the curiosity to see something happening. |
| 20 to 60 min | People stopping for medium time range in the area for longer purposes, such as having lunch, participating in some activities, going to the library or one of the bars and pubs present, etc. |
| 60 to 120 min | People stopping for a time range medium-long in the area. These people usually are doing some long activities, such as attending a conference or a lesson, enjoying time in a café or in a restaurant, participating in longer activities in the area. |
| 2 to 3 h | People stopping for a long time in the area, for example for attending a longer lesson or conference or event in the theatre or participating in quite long events; or just doing shorter activities in different spots inside the same area. |
| More than 3 h | People stopping for a very long time. Usually people that are resident, or work in the area, or they have longer classes or multiple classes in the same area, or they are spending their leisure time in the area for long time. |

According to the described types of analysis, we queried the platforms and we observed both graph and raw aggregated data, available in .csv format. The main steps of the methodology are highlighted in Figure 5.

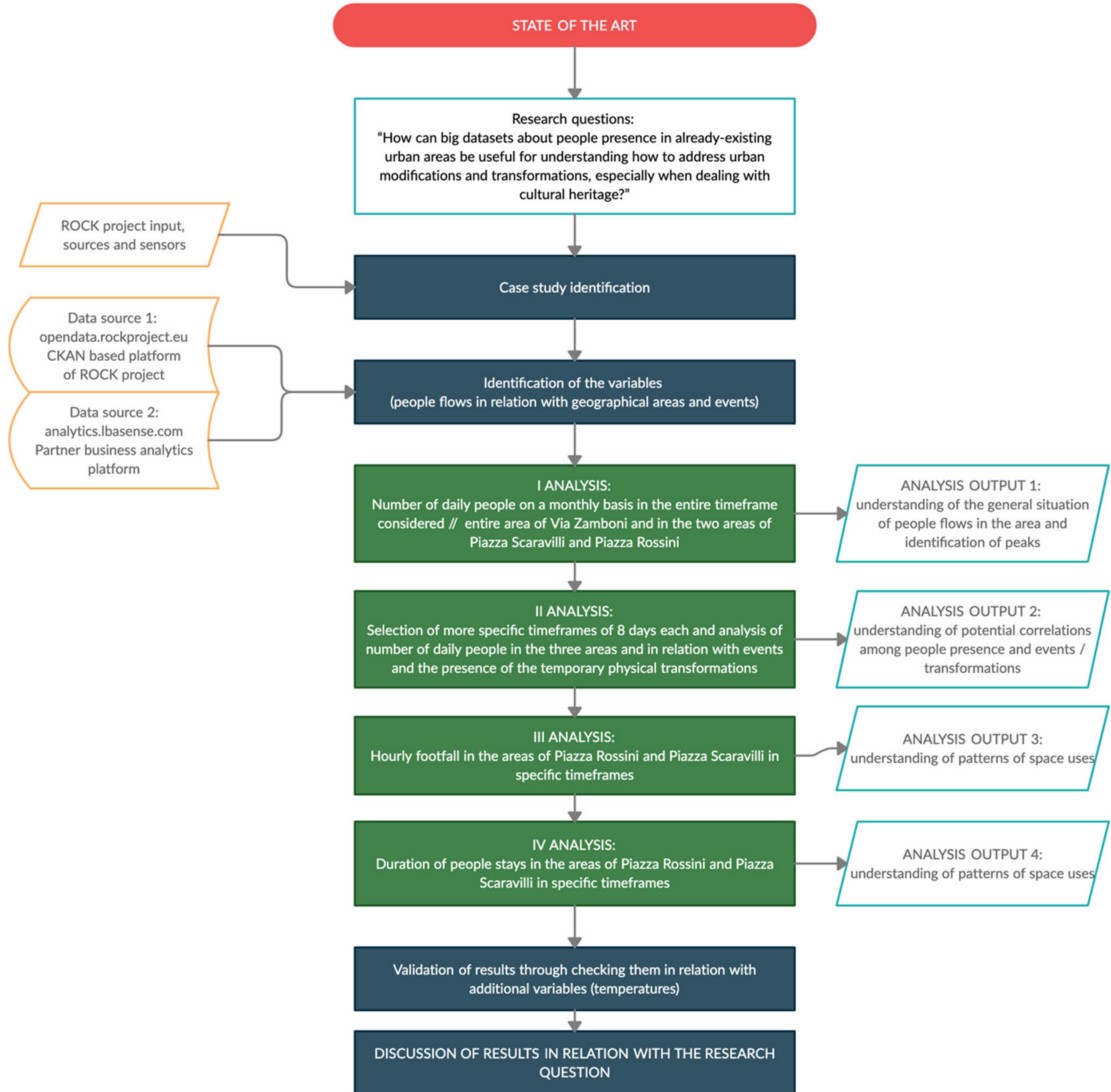

**Figure 5.** The steps of the methodology used for analyzing data.

### 3.2.1. Selection of Timeframes

There were two timeframes selected for analysis. The first one is the monthly period in which data were available, from June 2019 to September 2020. We were also able to include the first days of October 2020, but as the paper was submitted the 15th of October, we decided to stop the analysis with September.

The second one is a specific selection of eight days' timeframe characterized by a different presence of events and temporary transformation in the two squares of Rossini and Scaravilli. We selected eight days long timeframes in order to have a complete week of data, but with each of the timeframes including the complete weekend, so they all start on Monday or Tuesday. This is because most parts of events are in Italy organized during the weekends and it can be useful to also include in the same period the immediately following one or two days. The timeframes we selected are the following:

- 21–28 September 2020. In this timeframe both Piazza Scaravilli and Piazza Rossini hosted a temporary physical transformation, and, in addition, there was a series of events, under the name

of "Take Care of U". Especially located in Piazza Rossini, the events were expected to attract people to the area. It is important, in fact, to remember that Via Zamboni is a university street in the core center of the city of Bologna. Thus, it is frequented by a variety of people (students, tourists, businesspeople, residents, etc).

- 1–8 September 2020. In this timeframe, the area was characterized by the absence of events but the presence of both temporary installations in Piazza Scaravilli and Piazza Rossini.
- 4–11 October 2019. In this timeframe, the area had no events but both temporary transformation ongoing; specifically, in Piazza Rossini, there was the first pilot experience of greening named "Green Please: the meadow you don't expect" (Figure 3), built during the "Le Cinque Piazze".
- 23–30 September 2019. In this timeframe, in the area, there was the panel of events under the name "Le Cinque Piazze", where both the Screensaver and the first Piazza Rossini temporary installations were built.
- 10–17 June 2019. In this timeframe, the area was characterized by the absence of the temporary installations. The area was not involved in specific events, however, in the city, there was the annual summer panel of activities, that for 2019 went under the name of "BE HERE Bologna Estate 2019".

### 3.2.2. Events and Temporary Transformations in Relation with the Selected Timeframe

In order to take into account most parts of the boundary conditions in the analysis, we created a table including all the main events present in the area during the entire timeframe June 2019–September 2020. For creating this table, we referred to the news section of the ROCK project website (rockproject.eu), the news section on the ROCK Bologna website portal (bologna.rockproject.eu), and the Bologna cultural events website (agenda.comune.bologna.it). Table 2 shows the result of this overview. The table is divided into main events (with grey shading) and specific events inside the major one. We decided to include here only the most relevant sub-events (in white).

**Table 2.** List of main events in the area.

| Date | Hours | Event Title | Ref. Area |
|---|---|---|---|
| 18 August–30 September 2020 | | Take Care of U | Piazza Rossini |
| 29 September 2020 | From 17:30 | Presentation for the ROCK App "Bo for All" | Piazza Rossini |
| 26 September 2020 | 9:30–17:30 | Artistic marathon for Patrick Zaki | Piazza Rossini |
| 26 September 2020 + 8 September 2020 | From 17:30 | Take Care of U: "Ospiti alla Corte di Giovanni II Bentivoglio: una giornata tipo del 1492" | Piazza Rossini |
| | From 18:00 | Take Care of U readings/monologues with invited speakers (24 September 2020; 23 September 2020; 17 September 2020; 16 September 2020; 10 September 2020; 9 September 2020; 3 September 2020; 2 September 2020; 27 August 2020; 20 August 2020) | Piazza Rossini |
| 25 September–1 October 2020 | 9.00–18.00 | Installation and co-construction of "Green Please, il prato che non ti aspetti" | |
| 22 September 2020 + 25 August 2020 | From 18:00 | Take Care of U: "Amori, congiure e delitti: le famiglie nobili in Strà San Donato" | Piazza Rossini |
| 1 September 2020 + 18 August 2020 | From 18:00 | Take Care of U: "A passeggio con Rossini: personaggi illustri in Strada San Donato" | Piazza Rossini |
| 19 August 2020 | From 18:00 | Take Care of U: "Girovagando in via Zamboni osserviamo i palazzi, i portici, le piazze" | Piazza Rossini |
| 15 June–2 July 2020 | | Construction of Green Please | Piazza Rossini |
| 23–28 September 2020 | | Le Cinque Piazze | Via Zamboni |
| 25–28 September 2019 | 9.00–18.00 | Le Cinque Piazze: First installation of Green Please | Piazza Rossini |
| 23–28 September 2019 | 9.00–18.00 | Le Cinque Piazze: Screensaver installation | Piazza Scaravilli |
| 23–28 September 2019 | 9.00–18.00 | Le Cinque Piazze: UGarden | Piazza Verdi |

**Table 2.** *Cont.*

| Date | Hours | Event Title | Ref. Area |
|---|---|---|---|
| 23–28 September 2019 | 9.00–18.00 | Le Cinque Piazze: Mediterranean Design Weeks | Piazza Verdi |
| 13 May–30 September 2019 | | BE HERE—Bologna summer panel of events 2019 | City center |
| 28 August 2019 | From 20:30 | Performance teatrale: "Figli di una cavalcata" | Piazza Verdi |
| 8 July 2019 | 17:00–20:00 | Pianeti solitari: mappatura esperienziale | Via Zamboni |
| 7 July 2019 | 15:00–17:00 | Pianeti solitari: mappatura esperienziale | Via Zamboni |
| 18 June 2019 | From 17:00 | Carotaggi: passeggiata insolita | Via Zamboni |
| 14 October 2018 | From 17:00 | Concerto per 5 Pianoforti | Piazza Scaravilli |
| 6 July 2018 | From 18:00 | SLAB co-construction opening | Piazza Scaravilli |
| 2–6 July 2018 | 9.00–18.00 | Co-construction workshop SLAB | Piazza Scaravilli |
| 11–16 June 2017 | 9.00–18.00 | Co-construction workshop Malerbe #2 | Piazza Scaravilli |

## 4. Results

### 4.1. Via Zamboni

The decision to start the analysis from Via Zamboni relied on the geographic and social nature of the area in itself. For the purpose of the research question and for the objective of ROCK project, it was important to compare the very specific situations of Piazza Scaravilli and Piazza Rossini with the entire area. The entire via Zamboni is in fact object of ROCK project and the specific actions implemented in each of the squares are coordinated and defined in relation with impacts on the entire area. Thus, analyzing only the two squares is not sufficient to understand the dynamics of people frequency and use of space.

Analyzing the data available for the entire area, it is interesting to note some key features (Table 3):

- The area saw monthly a high amount of people passing in the area and going outside it or staying. Except for the COVID-19 period, the average number of people passing in the area on a monthly basis is around 1.5 million.
- There was an increase of people in the area in the months of September, October, and November 2019. It is important to note that in these timeframes, as shown in Table 3 there were some events organized in the area and the building of both Scaravilli and Rossini temporary transformations.
- The hours of a major presence in the area were the lunchtime hours (between 12:00 and 13:00), as the main timeframe during which presence was recorded. However, it was possible to record an increase of presence also during the first hours of the evening, starting from 18:00. These two observations can be explained probably with the nature of the area itself: lunchtime and aperitif time are naturally more crowded than the rest of the day when people are at work or inside the university buildings.

These general data, on a monthly basis, provide us with the overview and the base point for understanding the main trends but also for giving the framework for further analysis and considerations. However, it was necessary to refine the analysis in the specifically selected timeframes as a way to link precise periods of the year with the presence or absence of events and temporary transformations. Table 4 shows the result of the sum of unique visitors in the entire area of Via Zamboni in the selected timeframes, and it links these timeframes with the presence of events and physical transformations of space.

**Table 3.** Average and total daily visitors with hours of major presence for Via Zamboni and presence of transformations in the considered areas and major events. Please note that for events we consider here only major events, lasting more than 1 day.

| Timeframe | Sum Daily Visitors | Average Daily Visitors | Hours of Major Presence (Central European Time; CET) | Presence of Transformations // Major Events in the Area |
|---|---|---|---|---|
| 19 June | 1,628,414 | 54,280 | 12:00–13:00 | No // "Bologna Estate" set of events spread in the city center—no specific event in the area |
| 19 July | 1,797,050 | 57,969 | 12:00–13:00 (with slight difference with 17:00-19:00) | Yes // "Bologna Estate" set of events spread in the city center—no specific event in the area |
| 19 August | 1,111,597 | 35,858 | 12:00–13:00 and 18:00–19:00 | Yes // "Bologna Estate" set of events spread in the city center—performance in Piazza Verdi |
| 19 September | 2,327,649 | 77,588 | 12:00–13:00 | Yes // Yes (at the end of the month "Le Cinque Piazze" in the selected areas) |
| 19 October | 2,943,887 | 94,964 | 13:00 | Yes (both areas) // No |
| 19 November | 2,386,907 | 79,564 | 13:00 | Yes // No |
| 19 December | 1,749,274 | 56,428 | 13:00 (with slight difference with 16:00–18:00) | Yes // No |
| 20 January | 1,730,557 | 55,824 | 12:00–14:00 and 18:00 | Yes (COVID-19 emergency in Italy) |
| 20 February | 1,735,890 | 59,858 | 12:00–13:00 and 18:00 | Yes (COVID-19 emergency in Italy) |
| 20 March–June | No data | No data | No data | Yes (COVID-19 lockdown in Italy) |
| July (from the 17th) | 401,083 | 26,739 | 18:00–20:00 | Yes (slow re-opening from lockdown) |
| 20 August | 620,328 | 20,011 | 18:00–20:00 | Yes // Yes (at the end of the month "Take Care of U" in Piazza Rossini) |
| 20 September | 1,390,323 | 46,344 | 12:00–13:00 and 18:00–19:00 | Yes (both areas) // Yes ("Take Care of U" in Piazza Rossini) |

**Table 4.** Sum of unique visitors in relation to specific 8 days timeframes and link with events and physical temporary transformations.

| Timeframe (8 Days) | Unique Visitors (Sum) | Events | Transformation |
|---|---|---|---|
| 21–28 September 2020 | 431,516 | yes | yes |
| 1–8 September 2020 | 293,715 | no | yes |
| 4–11 October 2019 | 744,209 | no | yes |
| 23–30 September 2019 | 765,265 | yes | yes |
| 10–17 June 2019 | 448,059 | yes | no |

With some limitation in the possible interpretation (see Section 3.1.1), the table shows how the weeks with the presence of events were usually more crowded than weeks without events or without the temporary transformations. In particular, it is interesting to note the differences between June 2019 and September 2019. These two weeks are both weeks where events were presented and where university activities were still ongoing (for both timeframes, not a vacation period). Thus, it is possible to assume that both students and university staff were in the area. However, we see a very large difference (around 317,000 people) between the two timeframes giving us the opportunity to reflect on the following aspects:

- It is possible that the events in June and September were able to capture people in different ways, being, for example, the September event more attractive than the June one. As explained in Table 2, however, both months were involved into the annual summer panel of events, named for 2019 "BE HERE Bologna Estate". Nevertheless, 2019 saw the organization of a unique event in

September, organized by the ROCK project, under the name Le Cinque Piazze. In this event, both Piazza Scaravilli and Piazza Rossini temporary transformations were realized.

- The creation of the temporary transformations in both Piazza Scaravilli and Piazza Rossini could have influenced, at least partially, the use of those spaces that were both car parking before. However, it is not possible to say that they are responsible for the increase of people in the area, nor the still high presence in October because we were not able to compare those data with 2020 ones, due to the COVID-19 impacts.

- Finally, it is clear that the 2020 high decrease of people in the area is due to COVID-19 and to the consequent lockdown. However, the great increase of people between the first days of September and the end of the same month shows a progressive resumption in the use of public spaces. This increasing trend is also supported by Figure 6, where it is possible to see the increasing curve from the 31st of August to the 6th of October 2020. The peaks on 12th of September and starting from the 19th to the 28th can also be related with the presence of the panel of events "Take Care of U" in the area and, especially, in Piazza Rossini.

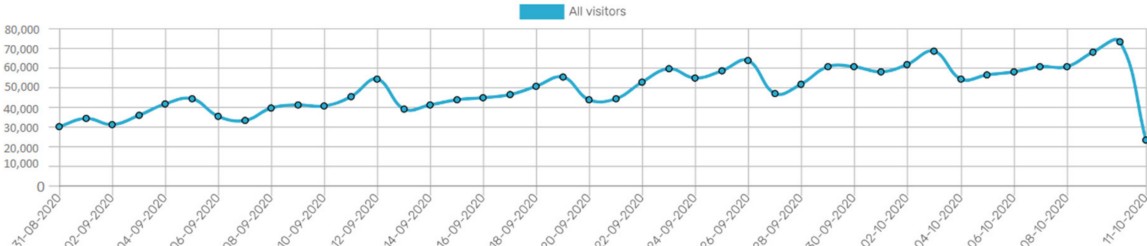

**Figure 6.** Increasing curve of visitors in the entire area after the COVID-19 lockdown. In particular, the graph shows the timeframe 31st of August to 6th of October.

In order to have a holistic view of the use of space inside the area, we performed an analysis in relation to the different segments of the area. The sensors, as shown in Figure 7, were distributed along via Zamboni, allowing us to understand if there were some specificities in the distribution of people along with it.

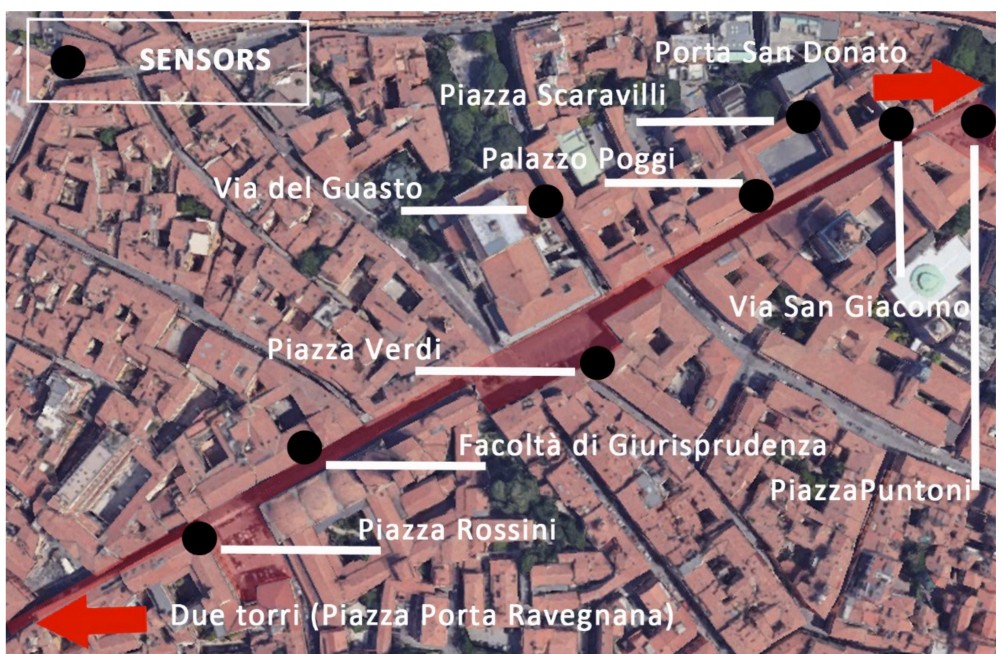

**Figure 7.** The sensors' positions in the area included between Piazza Rossini and Piazza Puntoni.

The following schemes (Figures 8–10) shows how there were some specificities:

- The sensors in Porta San Donato and the Two Towers always detected the most people. This is because these points are the limits of Via Zamboni, where people flow merges with other flows of the city. In particular, the area around Porta San Donato is particularly crowded as it is one of the ancient doors to the city center and it actually links it with the major communications avenues.
- Inside the area, Piazza Verdi is the most attended. It is, in fact, the major square where most part of the student's life outside happens, both during the day and the evening. It is also the area where nightlife usually concentrates, often creating tension with residents.

It is interesting to note how, in the graph showing the average distribution of people per area, related to the entire period June 2019–September 2020 (Figure 8), Piazza Rossini is in line with Piazza Verdi. These data can be explained considering a few aspects:

- The sensor was not located inside the square in a sheltered position, but it could detect people passing in front of it as well. In fact, for going to Piazza Verdi, this is one of the main flow directions, meaning that the number recorder for the Piazza Rossini sensor was not necessarily catching only people inside the area but also the ones passing in front of it.
- It is also possible that the progressive re-appropriation of uncommon spaces, such as Rossini and Scaravilli, and the growing attention that is paid to that area, are effectively increasing also the use of this specific space. Personal observations regularly made in the area tend to support this aspect, even if they cannot be considered, up to now, the proof that the physical transformation of Piazza Rossini is the unique reason.

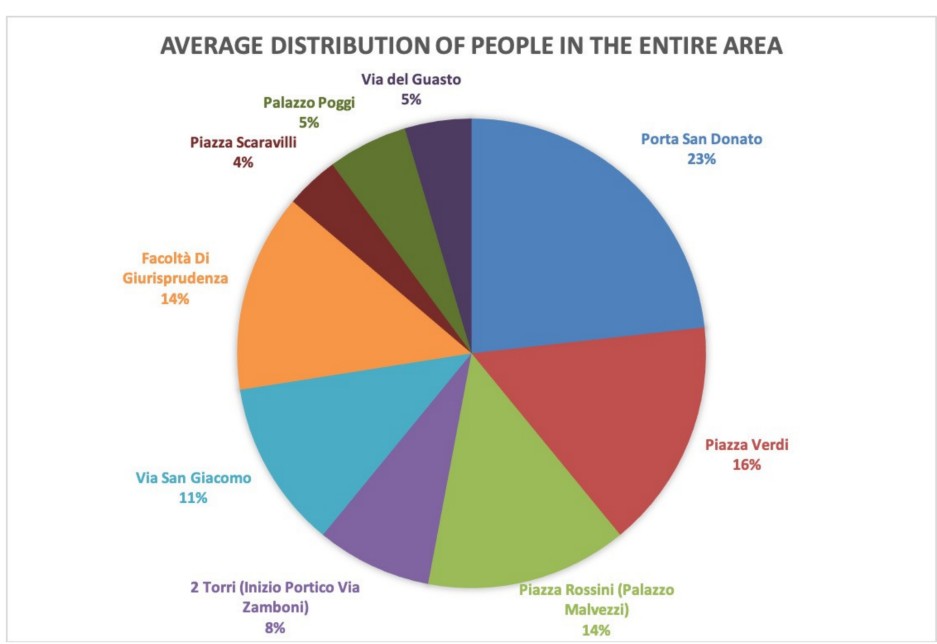

**Figure 8.** Average distribution of people in the entire area of Via Zamboni, for the timeframe June 2019–September 2020.

Indeed, the same graph analyzed for more limited periods of time, shows homogeneous results, with slight differences both for 2019 and 2020 (Figures 9a,b and 10a,b).

The final analysis we carried out for the entire Via Zamboni area was related to the average duration of people. These data are relevant as they show in-depth how people are used to exploring the area. Most people tend to stay in different spots for less than 5 min. This is easily explained with the nature of the area, which has been conceived for years as a transit to another destination. However, it is interesting to note that in all the timeframes with events or physical transformations, the number

of people staying longer (from 5 to 20 to 60 min) tended to increase. Conversely, the percentage of people staying longer than 1 h tended to be stable (Table 5).

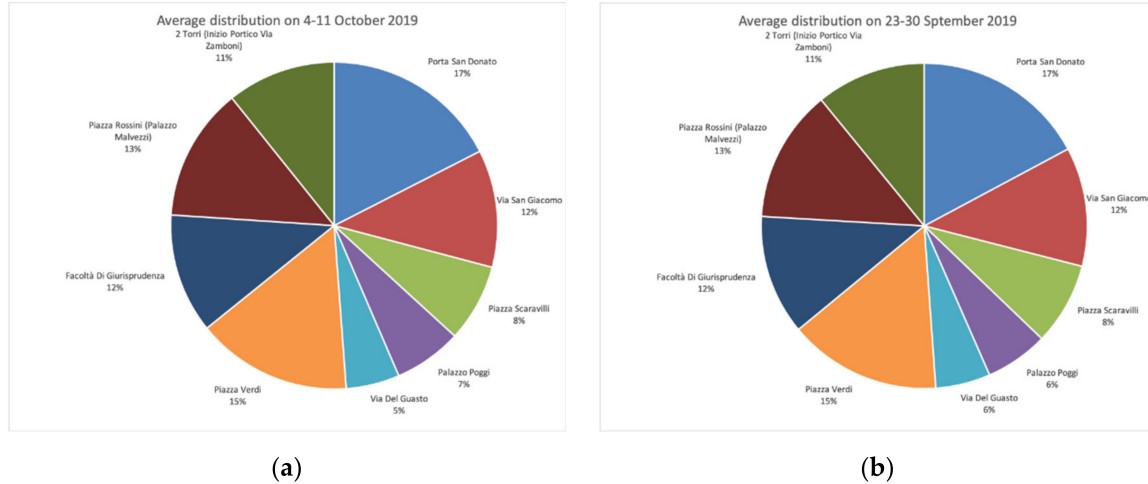

(**a**)                                                                                                          (**b**)

**Figure 9.** Average distribution of people in the entire area of Via Zamboni, for the timeframes 4–11 October 2019 (**a**) and 23–30 September (**b**).

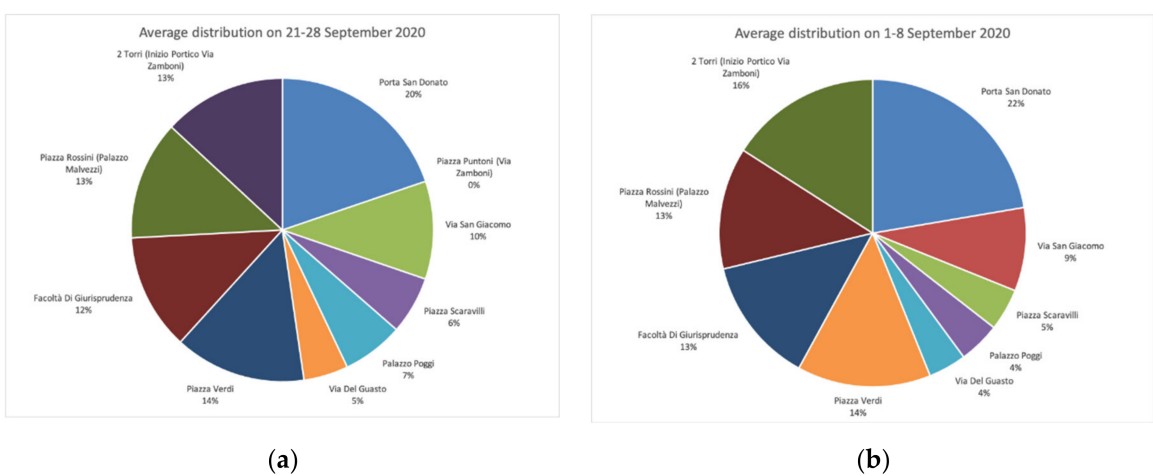

(**a**)                                                                                                          (**b**)

**Figure 10.** Average distribution of people in the entire area of Via Zamboni, for the timeframes 21–28 September 2020 (**a**) and 1–8 September (**b**).

**Table 5.** Duration of people stays in the entire area of Via Zamboni, in selected 8 day timeframes.

| Timeframe (8 Days) | <1 min | 1–5 min | 5–20 min | 20–60 min | 60–120 min | 2–3 h | >3 h | Events | Transformation |
|---|---|---|---|---|---|---|---|---|---|
| 21–28 September 2020 | 29% | 28% | 20% | 15% | 5% | 1% | 2% | yes | yes |
| 1–8 September 2020 | 30% | 28% | 19% | 14% | 5% | 2% | 2% | no | yes |
| 4–11 October 2019 | 31% | 27% | 20% | 13% | 5% | 1% | 2% | no | yes |
| 23–30 September 2019 | 23% | 27% | 24% | 18% | 6% | 1% | 1% | yes | yes |
| 10–17 June 2019 | 17% | 21% | 32% | 21% | 5% | 2% | 2% | yes | no |

These data show how, in general, people tended to be present and to stop when activities in the area occurred, even if their stay was relatively short. If, in particular, we consider the number of people present in the area in the two core timeframes of 2019 (the 10–17 and the 23–30 ones), it is possible to affirm that around 53% and 42%, respectively, stayed in the area from 5 to 60 min, corresponding to around 237,471 and 321,411 unique visitors. Parallelly, it is possible to observe that in two timeframes without events (4–11 October and 1–8 September), the shorter durations (less than 5 min) increased.

*4.2. Piazza Scaravilli*

As explained in the methodology section, we performed the same analysis for the three areas. Going now deeper into the specific area of Piazza Scaravilli, a few interesting points emerge in the analysis of monthly visitors.

It is important to say that the sensor located in this area was installed in a more protected part of the square relativethe others. On one hand, this feature tends to give lower results in term of the absolute number of people detected, on the other, it is more accurate in detecting mainly people that enter the square, and not the ones that pass near it.

Moreover, for that reason, the data analysis is very interesting and relevant in relation to the research question of this paper, as we have a clearer picture of the direct effects that the physical transformation is having and had in the past on how people use this space.

However, as addressed in Section 3.1.1, we need to express some limitations of the analysis that we are now able to do, as this specific sensor incurred in unexpected breakdowns, so there is a lack of data from the period starting in November 2019 to July 2020. Nevertheless, it is still possible to make some observations, especially in relation to the first five months (Tables 6 and 7):

- The months of June and July 2019 were almost aligned with more than 100,000 visitors, even if it is already possible to detect the progressive decrease that peaked in August 2020. These data are explainable in function of the academic year, as in Italy, August is a vacation period and in July many students leave the city. The month of July is, in fact, only an exam month, with lessons already finished. However, this time was also framed by the absence of temporary transformations in the area.
- In September and October 2019, there was a high increase in visitors. Partially these data can be justified with the restarting of lessons, but a cautious comparison with September 2020 (when lessons also restarted in person in Bologna) together with the amount of the increase in people can be probably better explained by the panel of events Le Cinque Piazze, which also built the actual physical transformation in the area. In particular, the installation was built at the end of September, thus the high presence of people in October was probably also influenced by the greening and the possibility to sit and have lunch outside.
- Table 7 shows also the confirmation that the square has been recently used with increasing time duration. The percentage of people staying from 5 to 60 min in the square is around 34% in the 21–28 September timeframe and 27% in the 1–8 September timeframe. We do not know if this can be related to the presence of the event in Piazza Rossini. Probably, in this case, it was more linked with the restarting of the academic year, which happens at the end of the month, rather than at the beginning. This year the first lessons, in fact, started on Monday 21 in several university departments.

An element that tends to confirm this type of use of space is the average hourly footfall, where it is possible to observe how the use of this space was mainly concentrated during the lunch break, confirming that it tends to be used by persons that are near this area for staying outside during lunch (probably mainly student and university staff) (Figure 11). Additionally, it is important to note that inside this square, there are no food and beverage retails, so the sensor can only detect people that are choosing to pass through the square not for buying food and drinks.

Figures 12 and 13 are further expressions of the bar chart, showing, for example, the two months of September and October 2019. In both graphs, in fact, it is possible to identify the peaks of space use during lunchtime and mainly during the weekdays, while during weekends there was a decrease of use. Then, we also note the peaks at the end of September, when the events under Le Cinque Piazze were organized, and the following days in October. This last graph seems to confirm how the transformation of the space produced an increase in daily use of it from people.

**Table 6.** Average and total daily visitors with hours of major presence for Piazza Scaravilli.

| Timeframe | Av. Daily Visitors (Total) | Sum Daily Visitors (Scaravilli) | Av. Daily Visitors (Scaravilli) | Main Ref. Hours |
|---|---|---|---|---|
| 19 June | 54,280 | 181,555 | 6052 | 13:00 |
| 19 July | 57,969 | 101,884 | 3287 | 12:00–13:00 |
| 19 August | 35,858 | 85,597 | 2761 | 12:00–13:00 |
| 19 September | 77,588 | 354,739 | 11,825 | 12:00–13:00 |
| 19 October | 94,964 | 508,214 | 16,394 | 13:00 |
| 19 November | 79,564 | Sensor offline | Sensor offline | Sensor offline |
| 19 December | 56,428 | Sensor offline | Sensor offline | Sensor offline |
| 20 January | 55,824 | Sensor offline | Sensor offline | Sensor offline |
| 20 February | 59,858 | Sensor offline | Sensor offline | Sensor offline |
| 20 March–June | No data | No data | No data | No data |
| July (from the 17th) | 26,739 | Sensor offline | Sensor offline | Sensor offline |
| 20 August | 20,011 | 47,836 | 1543 | 13:00 + 18:00 |
| 20 September | 46,344 | 152,825 | 5094 | 13:00 + 18:00 |

**Table 7.** Duration of people stays in the area of Piazza Scaravilli, in specific timeframes.

| Timeframe (8 Days) | <1 min | 1–5 min | 5–20 min | 20–60 min | 60–120 min | 2–3 h | >3 h |
|---|---|---|---|---|---|---|---|
| 21–28 September 2020 | 24% | 24% | 19% | 15% | 7% | 3% | 8% |
| 1–8 September 2020 | 36% | 33% | 16% | 11% | 2% | 1% | 1% |

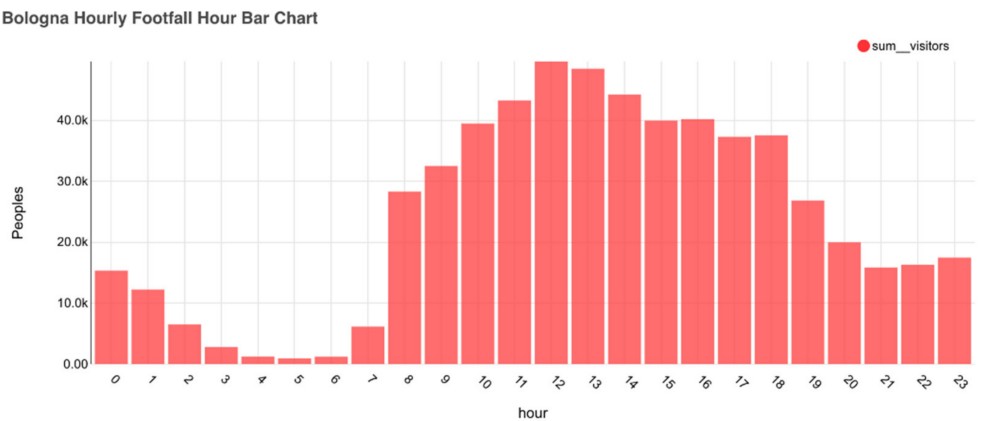

**Figure 11.** Piazza Scaravilli average hourly footfall bar chart divided per hour; month of October 2019.

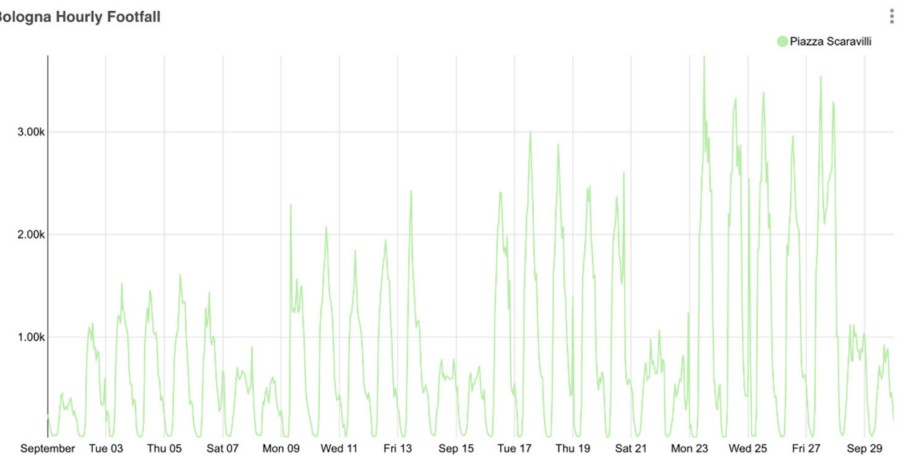

**Figure 12.** Piazza Scaravilli daily and hourly footfall; month of September 2019.

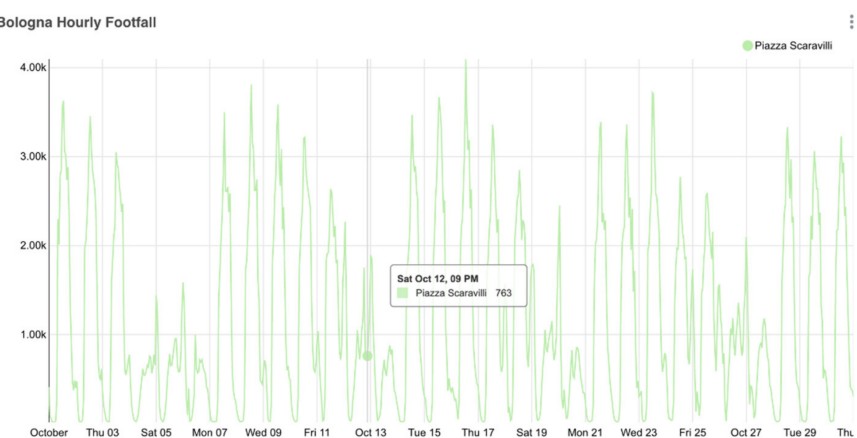

**Figure 13.** Piazza Scaravilli daily and hourly footfall; month of October 2019.

*4.3. Piazza Rossini*

The case of Piazza Rossini has homogeneous results with Piazza Scaravilli with some differences (Tables 8 and 9):

- At first, Piazza Rossini was more highly crowded than Piazza Scaravilli, but we already explained this fact also in relation to the geographical position of the square in a transit area and with the position of the sensor.
- As for Piazza Scaravilli, there was a high increase in the number of daily visitors starting from September 2019, in the occasion of the Le Cinque Piazze (Table 10). In this case, we also have the available data for November, showing how these three months in which the physical transformation and events were organized experienced a higher presence of people. However, as for Piazza Scaravilli, it is important to consider also that, being a university neighborhood, the amount of people tends naturally to be higher during the semesters (from September to July).
- The first temporary project for Piazza Rossini, in 2019, was successfully embraced by the citizens, with an average daily presence of around 25,000 visitors who did not just transit through the area but spent some time in the square. During the week of experimentation in September 2019, an increase in flows with an average of 20,000 daily footfall and a peak in the day of around 26,000 visitors was registered: the total weekly inflows amounted to 200,000 visitors.
- Finally, as for Piazza Scaravilli, here the bar chart and the graphs in Figures 14–18 also show the peaks in relation to the different hours of the day. In this case, we have daily a double-peak during lunchtime and during the evening, from 18:00.

**Table 8.** Average and total daily visitors with hours of major presence for Piazza Rossini.

| Timeframe | Av. Daily Visitors (Total) | Sum Daily Visitors (Rossini) | Av. Daily Visitors (Rossini) | Main Ref. Hours |
|---|---|---|---|---|
| 19 June | 54,280 | 366,429 | 12,214 | 12–13/18–22 |
| 19 July | 57,969 | 447,033 | 14,420 | 12–13/18–19 |
| 19 August | 35,858 | 241,676 | 7796 | 12–13/18–22 |
| 19 September | 77,588 | 618,953 | 20,632 | 12–13/18–19 |
| 19 October | 94,964 | 809,458 | 26,112 | 18:00 |
| 19 November | 79,564 | 687,372 | 22,912 | 13:00 |
| 19 December | 56,428 | 474,262 | 15,299 | 13:00 |
| 20 January | 55,824 | 437,087 | 14,111 | 12–13/18–19 |
| 20 February | 59,858 | 473,095 | 16,314 | 18:00 |
| 20 March–June | No data | No data | No data | No data |
| July (from the 17th) | 26,739 | 108,403 | 7227 | 22:00 |
| 20 August | 20,011 | 159,084 | 5132 | 12–13/18–19 |
| September 20 | 46,344 | 362,237 | 12,075 | 12–13/18–19 |

**Table 9.** Duration of people stays in the area of Piazza Rossini, in specific timeframes.

| Timeframe (8 Days) | <1 min | 1–5 min | 5–20 min | 20–60 min | 60–120 min | 2–3 h | >3 h |
|---|---|---|---|---|---|---|---|
| 21–28 September 2020 | 34% | 34% | 18% | 11% | 2% | 0% | 1% |
| 1–8 September 2020 | 32% | 34% | 18% | 12% | 2% | 1% | 1% |

**Table 10.** Average visitor stays in Piazza Verdi and Piazza Rossini and difference, in specific timeframes.

| Timeframe | Av. Daily Visitors (Total) | Av. Daily Visitors (Verdi) | Av. Daily Visitors (Rossini) | Difference |
|---|---|---|---|---|
| 19 June | 54,280 | 14,320 | 12,214 | 2106 |
| 19 July | 57,969 | 16,364 | 14,420 | 1944 |
| 19 August | 35,858 | 8,245 | 7,796 | 449 |
| 19 September | 77,588 | 23,354 | 20,632 | 2722 |
| 19 October | 94,964 | 30,660 | 26,112 | 4548 |
| 19 November | 79,564 | 26,760 | 22,912 | 3848 |
| 19 December | 56,428 | 17,450 | 15,299 | 2151 |
| 20 January | 55,824 | 15,999 | 14,111 | 1888 |
| 20 February | 59,858 | 18,631 | 16,314 | 2317 |
| 20 March–June | No data | No data | No data | No data |
| July (from the 17th) | 26,739 | 8530 | 7227 | 1303 |
| 20 August | 20,011 | 5611 | 5132 | 479 |
| 20 September | 46,344 | | 12,075 | |

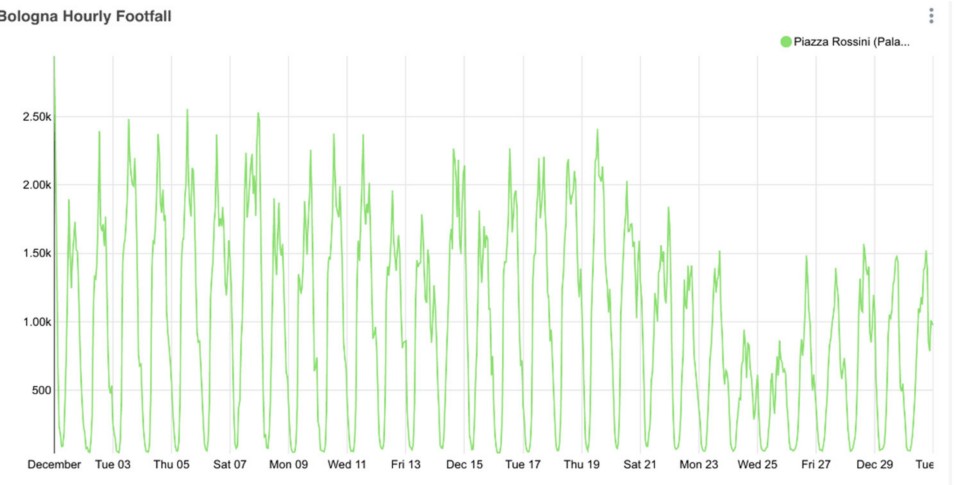

**Figure 14.** Piazza Rossini daily and hourly footfall; month of December 2019.

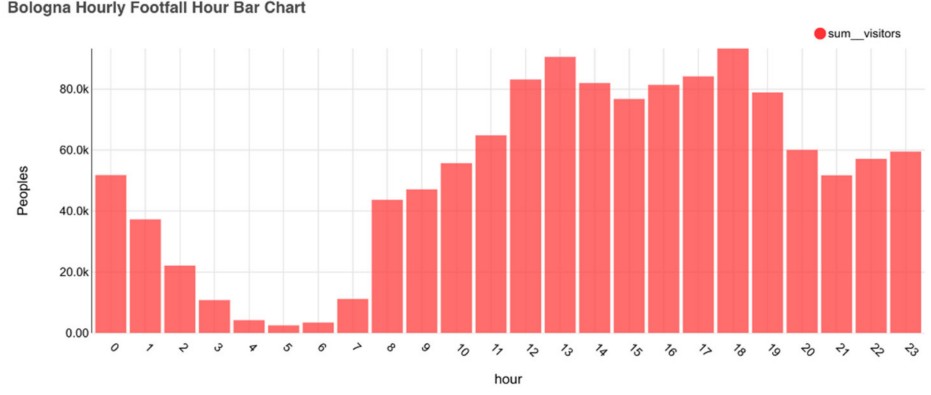

**Figure 15.** Piazza Rossini average hourly footfall bar chart divided per hour; month of October 2019.

In this case, Table 9 does not show a high increase in the duration of people stay for the two September 2020 timeframes. This is probably due to the fact that in the area there were some activities already from the 1st of September, as shown in Table 2.

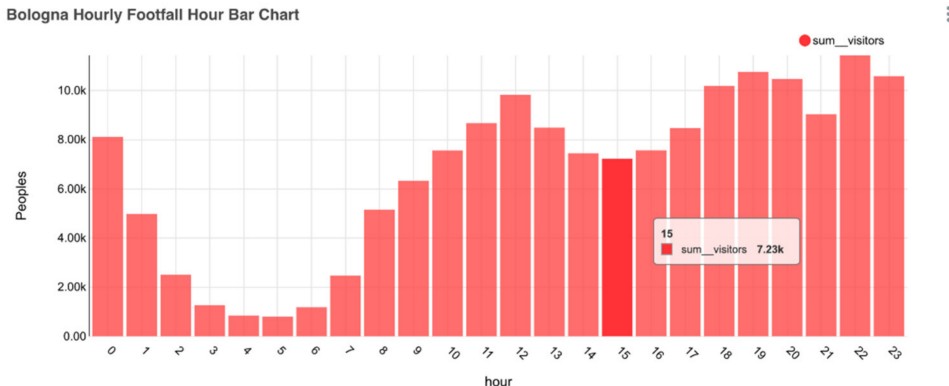

**Figure 16.** Piazza Rossini average hourly footfall bar chart divided per hour; month of July 2020.

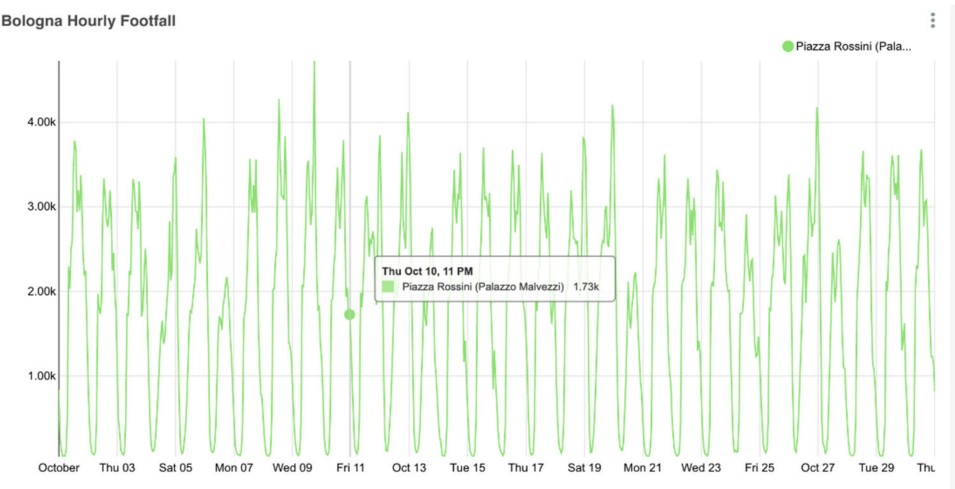

**Figure 17.** Piazza Rossini daily and hourly footfall; month of October 2019.

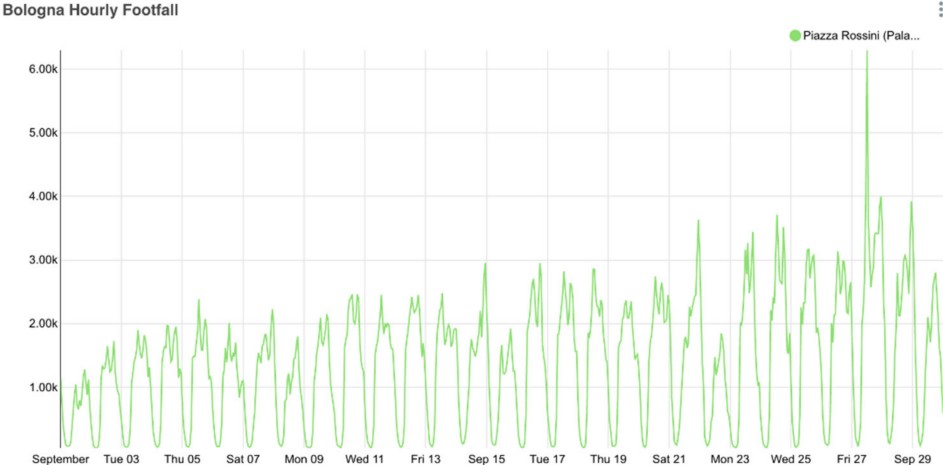

**Figure 18.** Piazza Rossini daily and hourly footfall; month of September 2019.

## 5. Discussion

The results coming from the data are readable in different ways. As anticipated in the introduction, in fact, the complexity of existing urban areas and, especially, areas framed by the high presence of cultural heritage and social issues cannot lead to a univocal interpretation of data. The multi-layered nature of these types of areas together with the potential proliferation of variables able to influence data cannot be completely considered, as many of the variables are unknown or non-predictable. We are not inside a controlled environment, thus we need to observe and read data trying to interpret them as best as possible, taking into consideration all these limitations, in order to avoid the so-called technological googles [27].

That considered, we can still give some interpretation of the results, as follows:

- It seems clear from the data shown that the presence of events can induce people to visit urban spaces more than usual.
- In the specific selected area and as the main attended events were organized for co-constructing and building the temporary installations, it is possible to argue that the physical transformation of these underused spaces has led to a major use of them.
- According to data, it is possible to say that the months following the construction of physical transformations tend to register a continuous presence of people in these areas, showing that they are really used by people.
- According to data, the two squares involved in this study are clearly used the most during the lunch break by people who must do their activities near the areas, as their duration of stay is always around 5 to 60 min. Probably, we can imagine that students and university staff, and potentially some residents, are the two major users of these spaces.
- The area is mainly frequented during the university semesters, starting from September and finishing around the end of June–July. The high decrease of attendance in August can thus be mainly reconducted to this reason. This element also confirms how the main users of these spaces are students or university staff.
- As a final verification, Figure 19 shows the correlation between temperatures and visitors from June 2019 to February 2020. The graph shows that there was no correlation between the seasons and how people use the space, explaining also that temporary transformations in this area can be useful for enhancing the use of spaces regardless of the temperature and the season. Further publications will deepen this aspect with the aim to better understanding the relations among microclimate and people.

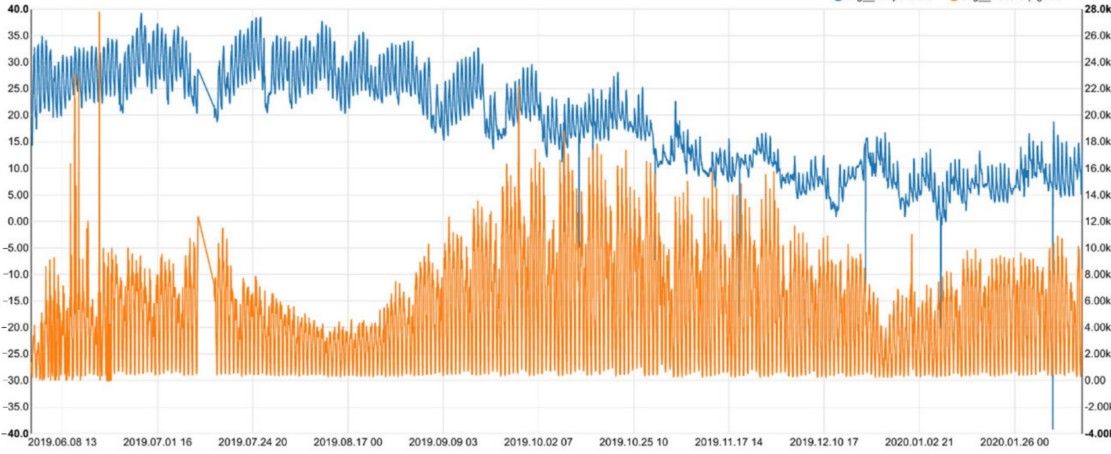

**Figure 19.** Relation between daily temperature and daily footfall in the entire Via Zamboni area for the timeframe June 2019–February 2020.

## 6. Conclusions

The present paper aims to demonstrate that innovative solutions for improving livability, sustainability, efficiency, resilience, accessibility, and equity in CH contexts may benefit from big dataset availability in order to facilitate the planning but also the verification phase of the real effects of the adopted measures.

In the case of Via Zamboni, Piazza Scaravilli, and Piazza Rossini, this affirmation is confirmed by the results coming from data and showing some interesting elements, such as the different durations of people stay, the role played by specific events in the areas and, finally, the most frequented hours during the day.

This last element, in particular, is also interesting in order to understand how to replicate this type of action in the same area but also in other areas, for example, considering that most people visit the area for small to medium amounts of time during lunch. The design of urban furniture and structures can, for example, take into consideration these data, by facilitating those types of activities.

Data were of high relevance in the case of Piazza Rossini. In fact, the Bologna administration decided to permanently transform the square from a car park to an open space for communities, due to the success and the number of people attending and visiting this space. We still do not know the architectural form that the place will have in the future, but the experience of the greening has been crucial in this decision. The results coming from this study can give some additional information about potential ways to permanently transform this space. This installation has also been particularly commented on and appreciated on social media from associations, social entrepreneurs (e.g., Salvaiciclisti, Dynamo, Kilowatt) and professional associations (e.g., the Bologna "Ordine degli Architetti"). Furthermore, citizens expressed their appreciation by signing the petition called "A meadow in Piazza Rossini". However, some critiques came from other citizens and professionals in the city, but the negative comments (mainly linked with the implementation of green actions instead of "stone ones") show the importance of these interventions to trigger dynamics of change, re-activation of communities, and discussions on the topic. In fact, this experience was also a way to discuss together the role of greenings in existing urban contexts, climate change, mitigation, and adaptation, and how a modern city dealing with tangible and intangible CH can imagine its future.

As a final consideration on the data we used to perform this analysis, it is important to note that only aggregated data were available, for ethical reasons. In fact, data collected by sensors should be able to give more information, for example on the trajectories of single people but those data were not made available in order to protect people's identities. We hope, in the future, to also have aggregated data on trajectories in a depersonalized way in order to understand the different flows and directions of people in the area.

Finally, in this period of uncertainty due to the COVID-19 emergency, data are also very useful in order to understand how spaces can be adapted for being safely used by citizens. Sensors able to detect the number of visitors and the duration of their stay can be relevant not for controlling but for addressing temporary transformations to allow people to use the space more safely. In the case of a university area, this topic is highly relevant as open spaces can constitute an extension of indoor spaces if correctly equipped, for a variety of activities: lunch breaks, lessons, meetings, relaxation spaces, etc. In the city of Bologna, where open spaces are also framed by the presence of porticoes, this reflection should be more exploited in order to really make the outdoor space an extension of the indoor one. Regarding this issue, there is still room for several innovations and big datasets can help to understand how to improve the use of those spaces.

**Author Contributions:** Conceptualization, S.O.M.B., R.R.; methodology, S.O.M.B., R.R.; validation, S.O.M.B., R.R., D.L.; formal analysis, S.O.M.B.; investigation, S.O.M.B., R.R.; data curation, S.O.M.B.; writing—original draft preparation, S.O.M.B., R.R.; writing—review and editing, R.R., D.L.; visualization, S.O.M.B., R.R.; supervision, D.L.; project administration, D.L.; funding acquisition, D.L. All authors have read and agreed to the published version of the manuscript.

**Funding:** This research was funded by ROCK-Regeneration and Optimization of Cultural heritage in creative and Knowledge cities, Grant Agreement number 730280.

**Conflicts of Interest:** The authors declare no conflict of interest.

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
