# Peer review of "Data Evidence-Based Transformative Actions in Historic Urban Context—The Bologna University Area Case Study"

_smartcities, doi:10.3390/smartcities3040069_

Round 1

Reviewer 1 Report

The  research and the paper has the potential to be contributing work to study crowd movement in urban spaces. However  how does collected data could be used in Cultural Heritage ( CH) context is not clear. Cultural heriatge is a very complex phenomenon . It associates complexities  of its connection with the people and communities with varied background  in terms of heritage meaning as well as place making. It is ( CH) further complicated  by its layered nature of historic connections. How does these  synchronic and diachronic connections would be measured via this crowd  movement data is not clear in this paper. No literature review on this aspect is not inluced  as well. The attempt of measusirng this via the data collected through sensor over simplified the phenomenon which might adversely affect the cultural heritage  (CH) context of the urban spaces. Moreove  the heritage overlay of place accumulates through ages. In this context on three years of intervention and the measured data  to assess or associate the heritage  aspects of the place.

Hence it was felt that the Cultural Heritage  (CH) context of the paper has little or no relevance. Based the collected data and the analysis it could be a very good paper on studying crowd movement in urban spaces and make relevant planning/ urban design  decisions. It is recommneded that the paper should be  revised by  shifting focus from Cultural Heriatge ( CH) to urban design or planning aspects  of the study.

Reviewer 2 Report

The article is interesting and under the scope of the journal. It is scientifically sound and well written. Results are useful in new studies. Only some improvements are advisable before publication:
1. Remove highlighting from bibliographic references [], Figures, and Tables.
2. Format errors, different margin to the left and right (see Line 362-363; 554-556)

Reviewer 3 Report

This paper analyzed human mobility data in historical cultural heritage site in Bologna. The authors used “ROCK City People Flow tool” which collect human mobility information by Large Crowd Monitoring sensors utilizing ICT (Information and Communication Technology).

Due to privacy concern, the only aggregated data are available for analysis. This is common problem for researchers in big data analytics especially in human mobility studies. Although the aggregated people flow data is limited to study in travel behaviors (unlike trajectory data which is movement information available by user or device ID), the authors effectively analyzed the data and applied in useful ways.

Overall, the written style and content arrangement is fine. I have only one comment in Figure 17. The authors said that “there is no correlation between the seasons and how people use the space”. If so, it should be included correlation coefficient value R. It looks some reverse correlation in first part of the graph. Please add x-axis label to see seasonal variation. I hope that the integration of other weather and environmental data will help to predict potential visitors.

Reviewer 4 Report

This work deals with Cultural Heritage and Smart Cities by referring to the project ROCK – “Regeneration and Optimization of cultural heritage in Creative and Knowledge cities ”.

The research topic could be interesting, but some revisions are needed in order to make the paper a suitable scientific work for a publication.

After giving it a thorough reading, I have some comments and I hope to provide both useful and helpful observations in order to enable the authors to improve their work. Please, see specific comments.

Abstract:

  • Add a sentence on evidences/results of the paper.
  • Keywords do not include the same words of the title.

In section 1 “Introduction”:

  • The introduction appears to be repetitive and dispersive in some places. Add/update literature review.
  • LINES 39-42: In the reference [1], indicate the page number in which it is possible to find the sentence reported in the text
  • LINES 43: Add some references to support the sentence related to a “variety of tools and techniques”.
  • LINES 78-80: Better explain the objective of the paper and highlight the advancement of knowledge
  • LINES 102-106: Since the authors talk about the use of technologies in the field of smart cities, I think it is appropriate to refer to more recent works than references [9-15]
  • LINES 122-123: “Both in Smart Cities and CH studies, citizens and communities are perceived as core parts of the discussion for addressing changes in already existing urban contest”. This concept does not emerge in relation to CH compared to the literature cited previously. Consider implementing the literature review in this direction.
  • LINES 191-198: In this sentence the content of the work is better outlined. However, I believe that this should be clarified from the outset.

In section 2 “Materials and Methods”.  

  • LINES 202-206: Once again the goal of the ROCK project is explained. But this is the methodological section.Therefore, enter this information in the previous section.
  • LINES 207-221: The "research-action-research" approach used is clear. However, the detailed description in the following lines is quite generic and the aspects strictly related to the presented work do not emerge.While maintaining a methodological description, this effort is required of the authors.
  • I suggest the authors to create a real "methodological framework" with a block diagram, in which the procedural steps, analyzes, tools and variables considered are highlighted.

In section 2.2. “Case study selection”:

  • Since the ROCK project refers to three Replicator Cities, in this section highlight the characteristics in common and not between the three case studies, in order to understand the reason for the choice of the city of Bologna, not to be connected only with the origin and knowledge of the
  • Add information about related traffic data conditions in the analyzed area.
  • Insert a figure with the evidence of the street and the square under study with respect to the urban area.

In section 2.3. “The use of sensors in the ROCK project in Bologna”:

  • LINES 331-339: How and according to what criteria the parameters/quantities to be detected/measured have been chosen?

In Section 2.3.1 “Limitations on the sensors in the area”:

  • LINES 374-385: I believe that the time limitation is the greatest criticality of the work, due to the fact that there is no comparison between before & after data

In Section 3 “Results”:

  • LINES 462-463: Correct the italic font
  • LINE 488: Add a space after Table 3
  • Having the events in one table (Table 2) and the values of the daily visitors in another table (Table 3) does not allow immediate interpretation of the results. Evaluate this aspect. Partially made in Table 4,albeit in an aggregate way.
  • LINES 497: Correct error (i.e. Section 2.XX)
  • Figure 5: Why the value in correspondence of 11.10.2020 is so low?
  • Figures 6-7-8: Without a suitable map/figure for the location of the various points of interest, these graphs are difficult to interpret for any non-expert reader of the study area.
  • Table 5: Interesting analysis if well correlated also with the presence of events
  • Same consideration for next sub-sections.

Round 2

Reviewer 1 Report

The authors trie dto explain their position on CH in the latest version.

That provides a better idea about hwere CH sits within the scope of the paper. 

This is imporatnt to get the right kinds of reader, who might be benefitted from the  study.